# Development of Microwave Slow-Wave Comb Applicators for Soil Treatment at Frequencies 2.45 and 0.922 GHz (Theory, Design, and Experimental Study)

**Graham Brodie** [1,*] [ID]**, Yuriy Pchelnikov** [2] **and Grigory Torgovnikov** [3]

1   Faculty of Veterinary and Agricultural Sciences, Dookie Campus, The University of Melbourne, 940 Nalinga Rd., Victoria 3647, Australia

2   PchelnikovCosulting.Slow-Wave Structures Application, 104 Drexelbrook Ct, Cary, NC 27519, USA; yupchel@gmail.com

3   School of Ecosystem and Forest Sciences, The University of Melbourne, 4 Water St., Creswick Victoria 3363, Australia; grigori@unimelb.edu.au

*   Correspondence: grahamb@unimelb.edu.au; Tel.: +61-3-5833-9273

**Abstract:** In agriculture and industry, it is often necessary to heat surface layers of material like soil, timber, concrete, and so on, with microwave (MW) energy. Traditional MW irradiators (antennas) cannot restrain their heating to the surface, with the energy penetrating deeply into the material. Slow-wave comb applicators can provide the required energy distribution in the surface layer. Theoretical analyses of the comb applicators used for heating were carried out and on this basis, three comb applicators were designed and made for soil treatment: two applicators Comb 1 and Comb 2 for frequency 2.45 GHz and Comb 3 for frequency 0.922 GHz. An experimental study of applicators was carried out using two MW plants: 30 kW (2.45 GHz) and 60 kW (0.922 GHz) for heating soil with moisture content in the range from 32% to 173% and density 460 to 1290 kg m$^{-3}$. The study showed that comb applicators provide the following advantages: reduction in energy dissipation in material depth and release of the significant part of applied MW energy in layers close to the applicator surface. Comb applicators can provide the required soil top layer treatment (sterilization) with reasonable efficiency and can be recommended for practical use in shallow soil treatment for weed seed and pathogen control in agricultural applications. Comb applicators can also be used for effective heating and MW treatment of the surface layers of wood, concrete, bricks, plastics, and other dielectric materials.

**Keywords:** microwave; soil; heating; slow-wave applicator

## 1. Introduction

It has been demonstrated that microwave soil heating can deactivate weed seeds [1,2] and some soil borne organisms [3,4] in soil; therefore, microwave technology is being developed as a potential weed control strategy [5–7]. In many cases, especially in zero till cropping systems, the soil seed bank is within 2 cm of the soil surface [8,9]; therefore, to minimise energy requirements, it is necessary to only heat the surface layers with microwaves (MW). Traditional MW irradiators (antennas) cannot restrict their heating effect to only the surface layers and energy penetrates deeply into the material where the MW fields decay exponentially, according to Maxwell's equations describing propagation in a semi-infinite lossy medium [10], as energy is naturally absorbed by the material. The resulting energy wastage due to MW field transmission beyond the required heating depth (for example, if the heating depth in soil for killing weed seeds is only 10–20 mm), is very significant. Therefore, development of special MW irradiators for surface treatment, to increase process efficiency, is required.

To confine the microwave fields to the surface layers of soil, the slow-wave comb (SWC) structure, which is often are called a "surface wave" structure, was studied. Slow-wave structures have a characteristic comb structure (Figure 1), which concentrates the microwave field near the comb surface. Previously, slow-wave structures (SWS) were used mostly as delay lines [11] and as interaction circuits in MW vacuum devices, and their properties were explored for these specific applications [12]. The novel properties of slow-wave structures have prompted their use in novel technologies for industrial, medical, domestic, plasma generation, and military applications [13,14].

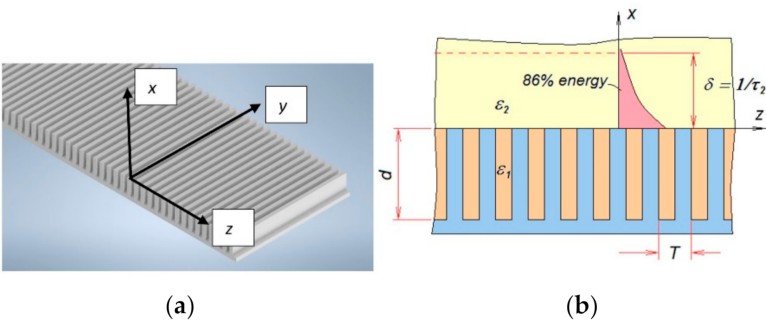

**(a)**                    **(b)**

**Figure 1.** Geometry of a comb-like slow-wave structure (**a**) showing the coordinate system and (**b**) the key dimensions of the structure.

Slow-wave structures have been used for heating thin dielectric materials since the 1960s because of their very high field intensity near the comb structure [15,16]; however, they are not commonly used for heating applications, because of the very limited thickness of material that the slow-wave fields interact with. More recently, interest in soil treatment for the destruction of weed seeds and plants has developed. As mentioned earlier, it has been demonstrated that more than 90% of the weed seed bank is within the top 2 cm of the soil profile in no-till cropping systems, like those practiced in Australia [9]. It has also been shown that conventional MW applicators used for weed treatment, such as horn antennas, project microwave energy to more than 10 cm into the soil [17]; therefore the research objectives of this work were to: model the energy distribution in the soil from slow-wave comb (SWC) structures; design and manufacture slow-wave structure applicators for soil treatment; experimentally study the heat distribution created by slow-wave comb applicators in soil; and provide recommendations for practical use of slow-wave applicators for weed seed bank treatment.

*Theoretical Analyses of a Slow-Wave Structure Used for Heating*

Slow-wave structures are non-radiating open transmission lines that have been used in charged particle accelerators and travelling wave tubes for more than half a century [18]. By their nature, slow-wave applicators confine the electromagnetic field distribution so that it remains very close to the surface of the slow-wave structure. The main idea of slow-wave propagation is that in periodic resonant cavities, such as a comb structure, the group velocity of the electromagnetic wave is decreased proportionally to the fineness of the cavity and consequently, the intensity of the propagating field is increased to conserve the energy flux [19]; therefore the field strength at the surface of the slow-wave structure is very intense and decays exponentially from the structure's surface, even in the absence of an absorbing dielectric material.

For a slow-wave comb structure, such as that shown in Figure 1, the microwave field, perpendicular to the slow-wave structure, is given by:

$$E = E_o e^{-\tau x} \tag{1}$$

where $\tau = k\kappa' \tan(kd)$, k is the wave number in free space, $\kappa'$ is the dielectric constant of the space above the applicator's comb $(\varepsilon_2 = \kappa' \times \varepsilon_0)$, and d is the depth of the comb structure (m).

If the applicator is protected by a dielectric plate (Figure 2) with a dielectric constant $\kappa'_1$, and the medium to be heated has a dielectric constant of $\kappa'_2$, then the field dispersion in the second medium is described by:

$$\tau_2 = -k^2\Psi_\tau \frac{\kappa'_1}{2\tau_1} + \sqrt{\left(k^2\Psi_\tau \frac{\kappa'_1}{2\tau_1}\right)^2 + \tau_1{}^2 + k^2\kappa'_1} \qquad (2)$$

where $\Psi_\tau = \frac{\tau_1\kappa'_2}{\tau_2\kappa'_1}$ and the subscripts correspond to the two materials. Note: due to the impedance transforming effect of the dielectric plate, $\tau_1 \neq k\kappa' \tan(kd)$, therefore:

$$\tau_1 = \sqrt{\tau_2{}^2 + k^2\left(\kappa'_2 - \kappa'_1\right)} \qquad (3)$$

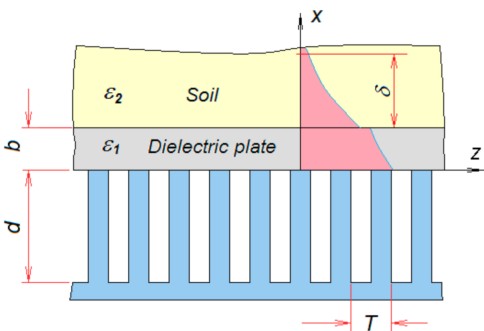

**Figure 2.** Slow-wave structure with a dielectric plate and soil adjacent to the structure.

The value of $\tau_2$ needs to be solved iteratively, using Equations (2) and (3).

The distribution of electromagnetic energy across the face of the applicator will be determined by the $TE_{10}$ mode excitation from the feeding wave guide and should be of the form:

$$E = E_o sin\left(\frac{\pi y}{A}\right) \qquad (4)$$

The temperature distribution is proportional to the square of the microwave field distribution and is determined from this squared field distribution by integration with $\frac{e^{-\frac{\xi^2}{4\gamma t}}}{\sqrt{t}}$, as outlined by Crank [20]:

$$T = E_o{}^2 \int_{-\infty}^{\infty} sin^2\left(\frac{\pi\xi}{A}\right) \times \frac{e^{-\frac{\xi^2}{4\gamma t}}}{\sqrt{t}} \times d\xi \qquad (5)$$

The resulting temperature distribution is described by a Voigt Function $U(y,t) = \Re\left\{\frac{e^{v^2}}{8t\sqrt{\gamma}} \times erfc(v)\right\}$, where $v = \frac{1 - \frac{j\pi y}{A}}{2\sqrt{\gamma t}}$.

It is useful to note that $\lim_{t \to 0} U(y,t) = \frac{1}{1 + \left(\frac{\pi y}{A}\right)^2}$.

It has been demonstrated [21] that the temperature due to microwave heating in a semi-infinite solid, like soil, is given by:

$$T = \frac{n\omega\varepsilon_o\kappa'' E_o{}^2}{8ka^2}(e^{4\gamma\alpha^2 t} - 1)\left[e^{-2\alpha z} + \left(\frac{h}{k} + 2\alpha\right)x \times \frac{-z^2}{e^{4\gamma t}}\right] + T_o \qquad (6)$$

The microwave heating problem is a thermal diffusion process; therefore, according to Holman [22], the multi-dimensional temperature distribution is determined by multiplying the

separate one-dimensional temperature distributions together. Therefore, the temperature distribution in the soil below the slow-wave applicator will be of the form:

$$T = \frac{n\omega\varepsilon_o\kappa'' E_o{}^2}{8ka^2}(e^{4\gamma\alpha^2 t} - 1)\left[e^{-2\alpha z} + (\frac{h}{k} + 2\alpha)x \times \frac{-z^2}{e^{4\gamma t}}\right] \times U(y,t) \times e^{-2\tau_2 x} + T_o \qquad (7)$$

The nomenclature of these equations is laid out in Table 1.

**Table 1.** Nomenclature used in the equations.

| Symbol | Meaning |
|--------|---------|
| E | Electric field strength of the electromagnetic wave (V m$^{-1}$) |
| Eo | Reference electric field strength of the electromagnetic wave (V m$^{-1}$) |
| $\tau$ | Evanescent field decay rate perpendicular to the slow-wave structure (m$^{-1}$) |
| x | Perpendicular distance from the surface of the slow-wave structure (m) |
| k | Wave number for the electromagnetic wave (m$^{-1}$) |
| d | Depth of the teeth on the slow-wave comb (m) |
| $\varepsilon$ | Dielectric permittivity of a material |
| $\varepsilon_o$ | Permittivity of free space |
| $\kappa'$ | Dielectric constant for a material |
| $y$ | The transverse dimension across the face of the slow-wave structure (m) |
| $\delta$ | Penetration depth of the microwave field (m) |
| A | Width of the slow-wave structure (m) |
| $\gamma$ | Thermal diffusivity of the heated material, allowing for simultaneous heat and moisture movement [21,23] (m$^2$ s$^{-1}$) |
| t | Time (s) |
| n | Scaling factor to account for simultaneous heat and moisture movement in heated material [21,23] |
| T | Period of the teeth on the comb structure (m) |
| $\omega$ | Angular frequency of the electromagnetic wave (Rad s$^{-1}$) |
| $\kappa''$ | Dielectric loss factor of the heated material |
| $\alpha$ | Electromagnetic field attenuation factor in the heated material (m$^{-1}$) |
| h | Convective surface heat transfer coefficient (W m$^{-2}$ °C$^{-1}$) |
| k | Thermal conductivity (W m$^{-1}$ °C$^{-1}$) |
| z | Longitudinal dimension along the surface of the slow-wave structure (m) |

## 2. Materials and Methods

### 2.1. Applicators Design

Based on the theoretical study outlined above, slow-wave comb (SWC) applicators for frequencies of 2.45 GHz (Comb 1 and Comb 2) and for 0.922 GHz (Comb 3), were designed. The SWC applicators, for both frequencies (2.45 and 0.922 GHz), are shown in Figure 3. The main dimensions of Comb 1, Comb 2, and Comb 3 are displayed in Table 2. The only difference between Comb 1 and Comb 2 are the length of the comb's electrode conic section, which helps transition between the field propagation in the wave-guide feed and the slow-wave structure, and the groove depth.

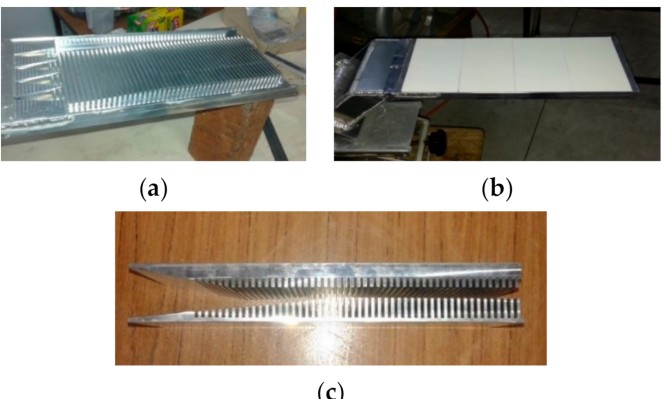

(a)                                    (b)

(c)

**Figure 3.** (**a**) Comb 2 applicator (2.45 GHz) without ceramic plates, (**b**) with ceramic plates (4 pieces of dimensions: 3 × 84 × 146 mm), and (**c**) comparison of Comb 1 and 2 electrodes.

**Table 2.** Applicator design parameters.

| Parameters | Comb 1 (2.45 GHz), mm | Comb 2 (2.45 GHz), mm | Comb 3 (0.922 GHz), mm |
|---|---|---|---|
| Working length | 356 | 356 | 346 |
| Applicator thickness | 23 | 23 | 37 |
| Applicator width | 150 | 150 | 264 |
| Comb electrode width | 100 | 100 | 150 |
| Comb electrode thickness | 16 | 16 | 28 |
| Comb electrode conic length | 90 | 185 | 120 |
| Groove depth | 6 | 13 | 13 |
| Groove width | 3 | 3 | 8 |
| Comb tooth thickness | 3 | 3 | 8 |
| Ceramic plates | Alumina (99%) ceramic plate size 3 × 84 × 146 mm (4 pieces), (Dielectric Constant = 9.8, loss tangent 0.0002) | | Alumina (99%) ceramic plate size 4 × 86.5 × 182 mm (4pieces), (Dielectric Constant = 9.8, loss tangent 0.0002) |

Note: The ceramic plates affect the microwave attenuation parameter $\tau_2$, which restricts the microwave heating to the upper layer of the soil. The thickness of the plates was chosen, based on variations in field distributions observed from electromagnetic simulation using the Finite-difference Time-domain simulation processes developed by Yee [24].

In the experiments, the surfaces of Combs 1 and 2 were covered by 3 mm thick alumina plates (Figure 3). The surface of Comb 3 was covered by 4 mm thick alumina plates (Figure 4).

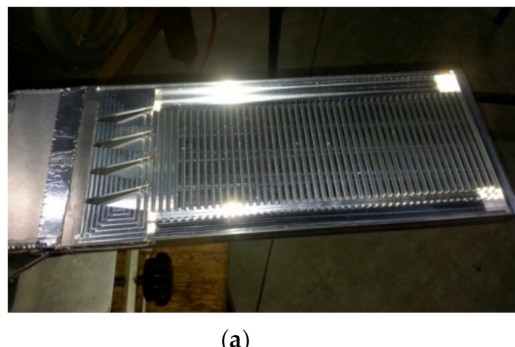

(a)

**Figure 4.** *Cont.*

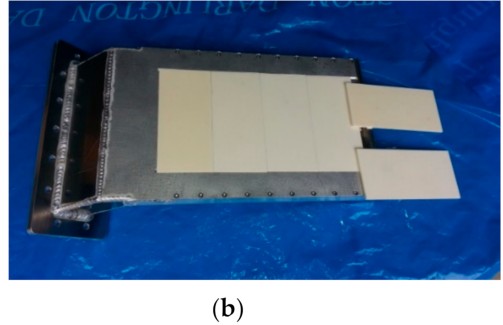

(**b**)

**Figure 4.** Comb 3 applicator (0.922 GHz) (**a**) without ceramic plates and (**b**) with ceramic plates (4 pieces of dimensions: $4 \times 86.5 \times 182$ mm).

## 2.2. Experimental Soil

"Potting Mix" was used as the test soil in these experiments (Figure 5). The soil had different moisture contents (MC), which ranged from 32% to 174% (dry weight basis), and different bulk densities, which ranged from 458 to 1290 kg m$^{-3}$. The soil used in the experiments had a significant percentage of organic particles of different sizes (wood, bark, grass); therefore, the dielectric parameters of the soil, at frequencies of 2.45 and 0.922 GHz, as a function of temperature, which ranged from 15 to 80 °C, had a dielectric constant ranging from 1.8 to 20 and a loss tangent from 0.2 to 0.5 [25,26]. The soil used in the experiments, which was a mixture of organic and mineral substances, also had different heat capacity and diffusion properties with high variability. Only moisture content and density of the soil were measured. The key parameters for the soil are outlined in Table 3.

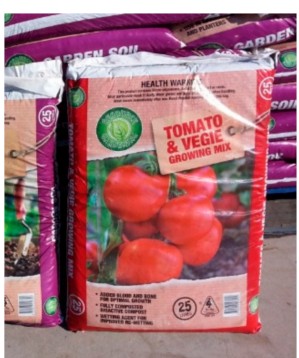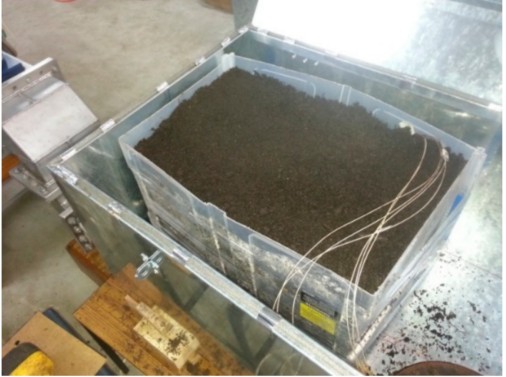

**Figure 5.** Soil for experiments in polypropylene container $310 \times 310 \times 550$ mm placed on the applicator in a metal leakage protection box.

**Table 3.** Summary of moisture contents and densities of soil mix used for each experimental assessment of the slow-wave combs.

|  | Moisture Content | Density (kg m$^{-3}$) |
|---|---|---|
| Comb 1—2.45 GHz | 42 | 458 |
|  | 173 | 806 |
| Comb 2—2.45 GHz | 32 | 586 |
|  | 89 | 710 |
|  | 174 | 1070 |
| Comb 3—0.922 GHz | 42 | 672 |
|  | 83 | 770 |
|  | 141 | 1290 |

The soil was placed into polypropylene containers, with dimensions of $120 \times 250 \times 300$ mm, for experiments at 2.45 GHz, and into containers, with dimensions of $310 \times 310 \times 550$ mm, for experiments at 0.922 GHz (Figure 5).

### 2.3. Experimental Installations and Procedure

A 30 kW (2.45 GHz) MW generator was used for experiments with the Comb 1 and Comb 2 applicators (Figure 6) and a 60 kW (0.922 GHz) MW generator was used for the experiments with the Comb 3 applicator (Figure 7). Auto tuners were placed in the MW wave guide systems to provide good matching of the generators and applicator (with soil), resulting in almost no MW power reflection. A circulator was included in the wave guide system to isolate and protect the magnetron. Figure 8 shows the scheme of the container with soil that was placed on an applicator.

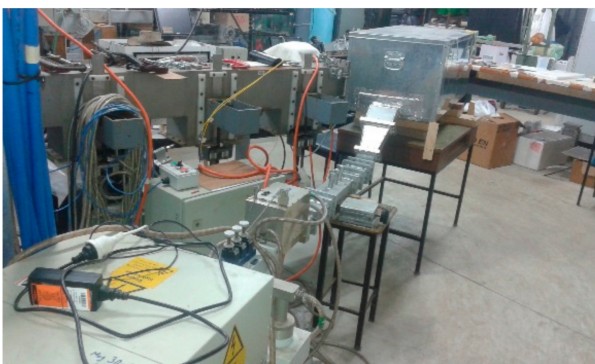

**Figure 6.** Comb 1 applicator (inside of metal box) connection with 30 kW MW generator 2.45 GHz.

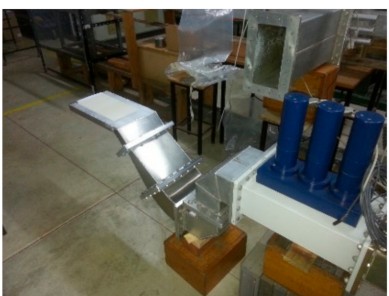 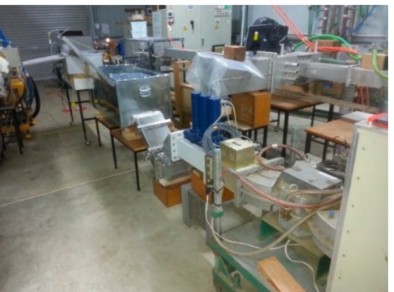

**Figure 7.** Comb 3 applicator (on the right-inside of metal box) connection with MW generator 60 kW, 0.922 GHz.

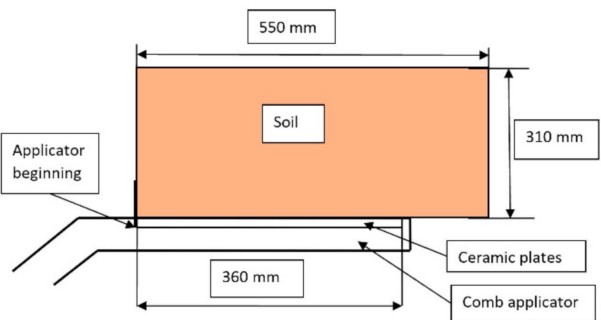

**Figure 8.** Comb applicator with soil in container.

Temperatures in the soil were measured at 490 points, using thermocouples, after each MW heating event was completed. Temperatures were measured at depths of 10, 30, 50, 80, 100, 120, and 140 mm at distances along the applicator of 30, 60, 90, 120, 150, 180, 210, 240, 270, and up to 395 mm from the beginning of the applicator, and across the applicator, at distances of 0.0, 37.5, 75, and 112.5 mm on both

sides of the centre plain of the applicator, across the comb structure. The distribution of measuring points covered a three-dimensional volume soil of 0.012 m$^3$ over the applicators.

Consistent temperature measurements were made by laying a jig onto the soil surface, allowing the thermocouples to be positioned accurately during measurements. The experimental conditions for each combination were repeated four times.

Values of MW power and time of the treatment were determined during preliminary tests to provide soil heating at a depth of 10 mm that was not more 80 °C, to prevent any opportunity for water boiling in the soil. After the container with the soil was placed onto the applicator and the leakage protection box was closed, MW energy was supplied to the applicators according to the following schedules: for applicators Comb 1 and Comb 2 (2.45 GHz), applied MW power was 3.5 kW, and treatment time was 15 s (providing an input energy of 53 kJ); and for applicator Comb 3 (0.922 GHz), applied MW applied power was 9–10 kW, and treatment time was 15 s (providing an input energy of 135–150 kJ).

Magnetrons can produce variable microwave power output, with variation between approximately 10% of their rated output and 100% of their rated output. Below 10% of the rated output power, the magnetron's output becomes less stable. In this case, the 2.45 GHz magnetron system was operating at 11.7% of its rated output; therefore, it should be operating within its stable range. The magnetron that was used in the experiment was a Model CWM-30S with an operating power range from 3 to 30 kW. The 60 kW, 0.922 GHz magnetron was operating at between 15% and 17% of its rated output, so it was also within its stable range.

We have assumed that the temperature distribution in the soil reflects the energy released in different locations within the soil and allows assessment of the energy distribution associated with each MW applicator.

## 3. Results and Discussion

### 3.1. Temperature Distribution in the Soil from the Comb 1 and Comb 2 Applicators (2.45 GHz)

Figure 9 shows a typical temperature distribution, at a depth of 10 mm, after applying 3.5 kW of MW power for 15 s.

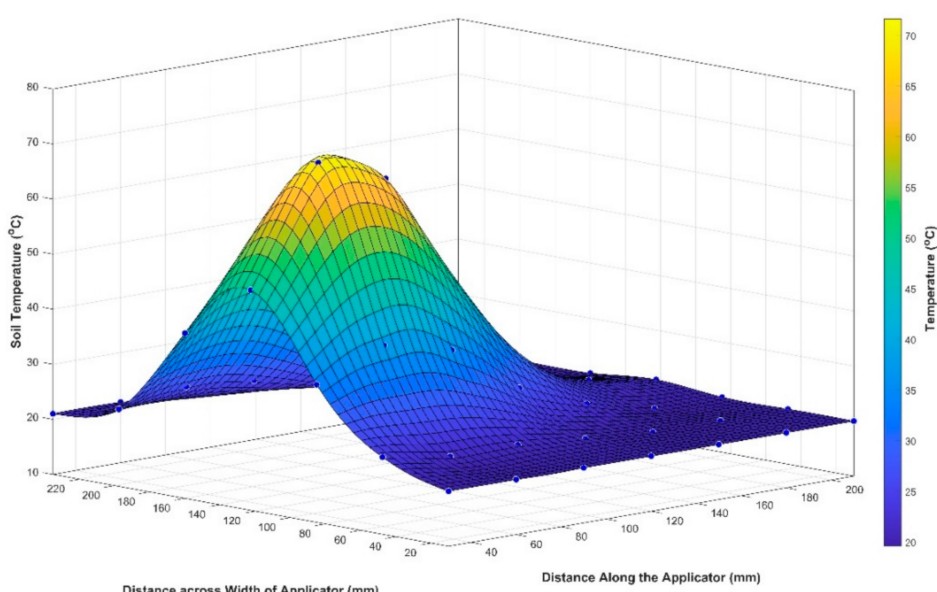

**Figure 9.** Temperature distribution in the soil on the depth of 10 mm by Comb 2 applicator at F = 2.45 GHz, P = 3.5 kW, time of MW heating—15 s, $T_0$ = 20 °C, applied energy 53 kJ. Soil moisture content MC = 89%, density 710 kg m$^{-3}$.

Both Comb 1 and Comb 2 gave very similar energy distributions along the applicator length. The maximum energy absorption takes place at about 60 mm from the beginning of the applicator. Figure 10 shows a comparison between the theoretically calculated temperature distribution, developed earlier in Equation (6), and the measured temperature in the soil, along the central plane of the Comb 1 applicator. The initial conditions were soil MC levels of 42% and 173%, initial temperature $T_o = 11$ °C, power P = 3.5 kW, and treatment time of 15 s. Almost all the energy was absorbed by 200 mm along the applicator under any soil moisture content regime. Most of the energy for soil at MC = 42% is absorbed in the top 50 mm of soil, while at MC = 173%, most of the energy is absorbed in the top 30 mm.

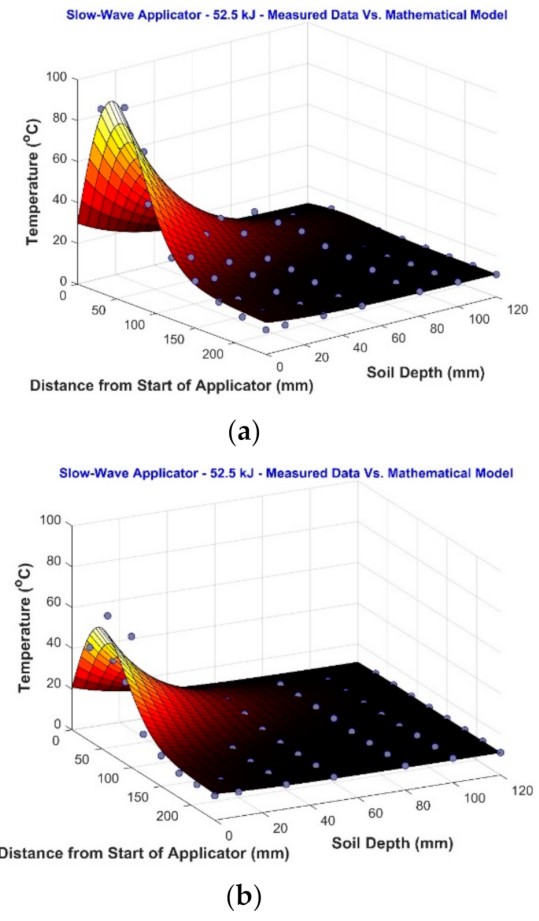

(**a**)

(**b**)

**Figure 10.** Temperature distribution in the soil along the Comb 1 applicator central plane, compared with the mathematical model in Equation (6), at F = 2.45 GHz, P = 3.5 kW, time of MW heating—15 s, $T_o = 11$ °C, applied energy 53 kJ. (**a**) MC = 42% and (**b**) MC = 173%.

Again, both applicators gave similar energy distributions across the applicator width. The temperature distribution across the Comb 2 applicator, at a depth of 10 mm and distances from the beginning of the applicator of 30, 60, and 90 mm, is displayed in Figure 11. The initial conditions of the soil are moisture contents of 32%, 89%, and 174%. Almost all the energy was absorbed within a width of about 150 mm across the applicator irrespective of soil moisture content.

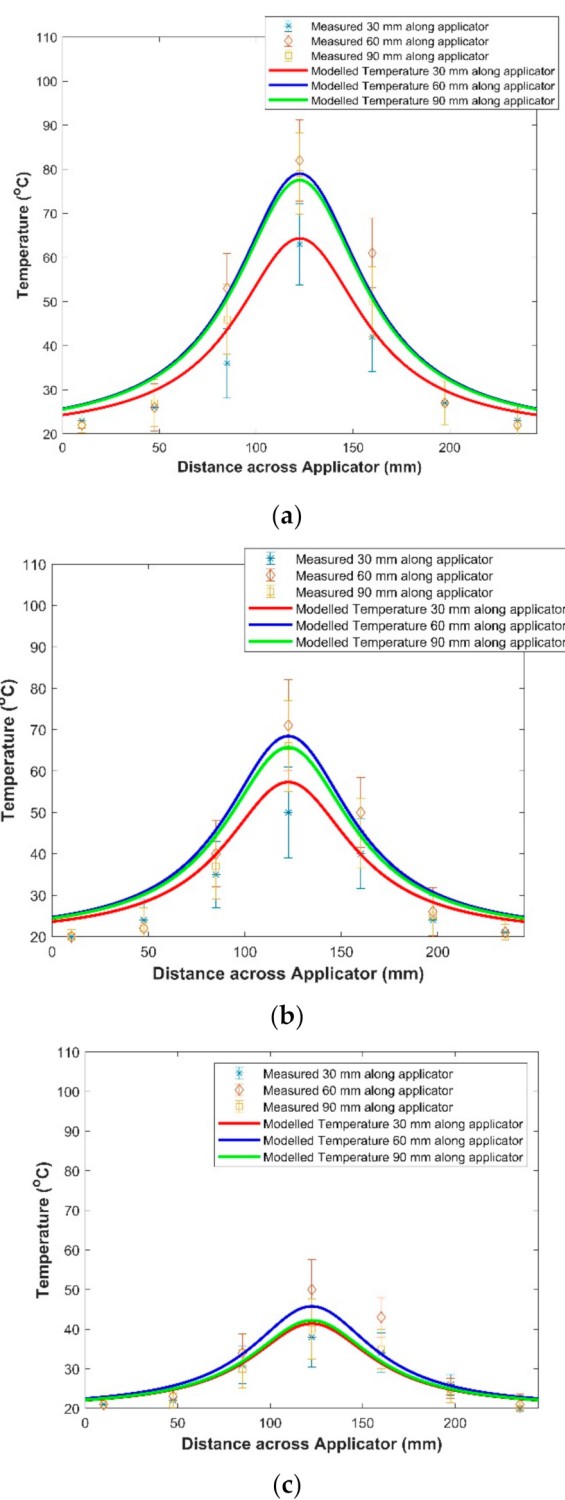

**Figure 11.** Temperature distribution in the soil after MW heating across the Comb 2 applicator at the depth 10 mm and distances 30, 60, and 90 mm from applicator beginning. (F = 2.45 GHz, P = 3.5 kW, time of MW heating 15 s, $T_o$ = 20–21 °C, applied energy 53 kJ), (**a**) MC = 32%, density = 586 kg m$^{-3}$, (**b**) MC = 89%, density = 710 kg m$^{-3}$, and (**c**) MC = 174%, density = 1070 kg m$^{-3}$. (Error bars represent the standard error for the measured data).

### 3.2. Energy Absorption as a Function of Soil Depth

Comparison of the temperature distributions in the soil, created by the Comb 1 and Comb 2 applicators, cannot be easily achieved because the soil that was used in the experiments had different

properties. For Comb 1, while the MC was 173%, the bulk density was 806 kg m$^{-3}$; however, when the same soil type was set up for Comb 2, the MC was 174% but the bulk density was 1070 kg m$^{-3}$. Some general comparisons will be made. Because the initial temperature of the soil in the experiments was 11 °C (Comb 1) and 21 °C (Comb 2), we have built graphs based on temperature increases (i.e., T–T$_o$). For example, for Comb 1, if the measured temperature at a point was 25 °C, the temperature increase was 25 − 11 = 14 °C. Figure 12 shows the temperature increases generated by applicators Comb 1 and Comb 2 in the vertical central plane at 60 mm from the beginning of the applicator.

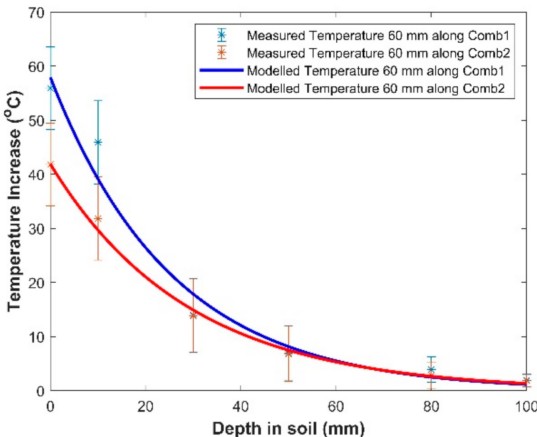

**Figure 12.** Comparison of temperature distribution on the soil depth by Comb 1 and Comb 2 applicators in vertical central plane at distance 60 mm from applicator beginning (area of maximum heating) (Error bars represent the standard error for the measured data).

Analysis shows slightly higher energy release between 10 and 20 mm depth from applicator Comb 1 compared with Comb 2. It can be explained by the shorter length of the conic part of the comb electrode. Both applicators give similar energy distribution along the applicator length. The maximum energy release in the soil takes place at 60 mm from the beginning of the applicator. Most of the energy was absorbed within 200 mm along the applicator for any soil moisture content. Both applicators provide similar energy distribution across the applicator width. Almost all energy is absorbed across the applicator width—about 150 mm. The conclusion is that Comb 1 and Comb 2 provide almost the same energy distribution in the soil.

The penetration depth for microwave energy from a horn antenna is directly dependent on the natural attenuation of microwave fields in soil ($\alpha$). In the case of the medium moisture soil used in this experiment, the natural attenuation of microwave fields with a frequency of 2.45 GHz is estimated to be 3.04 m$^{-1}$. The slow-wave applicator introduces a much more rapid attenuation of the microwave fields due to its evanescent field response ($\tau_2$). In the case of the medium moisture soil used in this experiment, the slow-wave attenuation of microwave fields with a frequency of 2.45 GHz is estimated to be 15.7 m$^{-1}$. This is 5.2 times faster field attenuation than would be expected from a horn antenna.

There is attenuation along the length of the comb applicator, which is in accordance with the natural attenuation of the microwave field in the soil ($\alpha$). This is to be expected and is acceptable because in a practical system, the applicator will be pulled slowly over the surface of the soil to provide more uniform treatment along a strip of soil. Although not being reported here, the strategy of moving the applicator has been proven to provide effective long-term control weeds and their seeds in the soil, with some longitudinal experiments showing weed-free conditions for more than 200 days (unpublished).

### 3.3. Temperature Distribution in the Soil by Comb 3 Applicators (0.922 GHz)

Figure 13 shows a typical temperature distribution in the soil at a depth of 10 mm after applying 9.5 kW of MW power for 15 s.

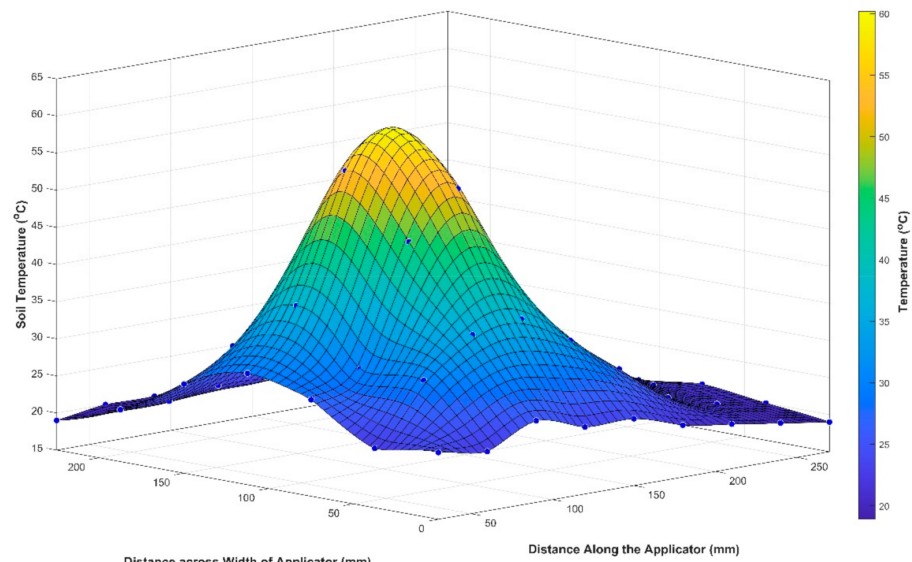

**Figure 13.** Temperature distribution in the soil at 10 mm depth by Comb 3 applicator at F = 0.922 GHz, P = 9 kW, time of MW heating—15 s, $T_o$ = 19 °C, applied energy 135 kJ. Soil moisture content MC = 142%, density 1290 kg m$^{-3}$.

The temperature distributions, along the central plane of Comb 3, compared with the mathematical model derived earlier, for soil MC of 42%, 83%, and 141%, are shown in Figure 14. The initial temperature ($T_o$) ranged from 15 to 20 °C and the applied MW power ranged from 9 to 10 kW (0.922 GHz).

These experiments showed that practically all the energy was absorbed by about 270 mm from the beginning of the applicator (Figure 14). The maximum energy release takes place at 100–120 mm from applicator beginning. During the 15 s of MW heating, the temperatures of soil surface reach 90–100 °C between 60 and 120 mm from applicator beginning. Wet soil temperature cannot rise above 100 °C because of the latent heat of vaporisation for water, which holds the temperature steady until the soil completely dries.

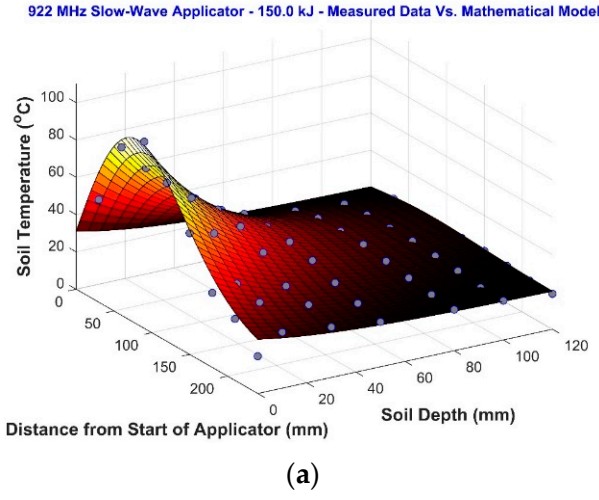

(**a**)

**Figure 14.** *Cont.*

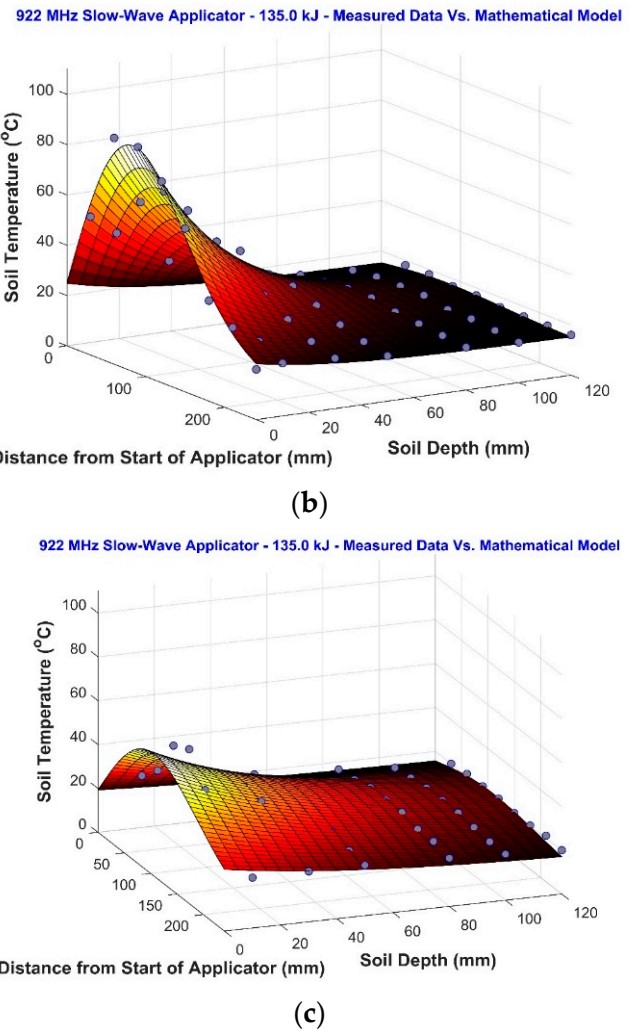

**Figure 14.** Temperature distribution in the soil, compared with the mathematical model in Equation (6), with different moisture contents by the Comb 3 applicator (0.922 GHz) along the applicator central plane on the depth 10 mm after MW heating during 15 s: (**a**) at MC = 42%—density 672 kg m$^{-3}$, T$_o$ = 17 °C, P = 10 kW, applied energy 150 kJ, (**b**) at MC = 83%—density 770 kg m$^{-3}$, T$_o$ = 15 °C, P = 9 kW, applied energy 135 kJ, and (**c**) at MC = 141%—density 1290 kg m$^{-3}$, T$_o$ = 20 °C, P = 9 kW, applied energy 135 kJ.

### 3.4. Temperature Distribution in the Soil Across Applicator

After applying 135–150 kJ of MW energy, most of the energy was absorbed in a width of 300 mm in the zones of maximum energy release (90 mm from the beginning of the applicator (Figure 15)). In all other zones, the energy distribution across the applicator was narrower. Most of the energy (up to 95%) irradiated by the Comb 3 applicator (comb electrode width 150 mm, applicator width 264 mm) was absorbed on the width of about 220 mm across the applicator and the difference in the soil moisture content does not significantly affect the energy distribution across the applicator.

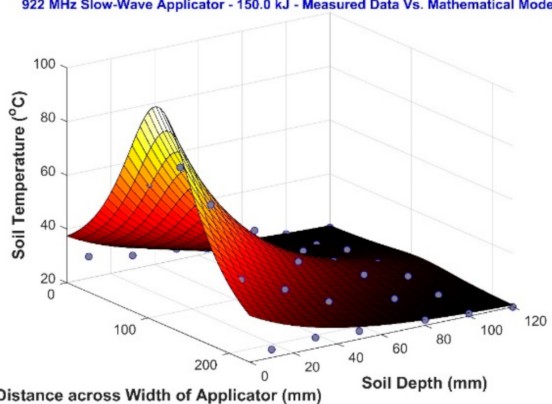

**Figure 15.** Temperature distribution across the applicator at different distances from applicator beginning, compared with the mathematical model in Equation (6), with MC = 42%. F = 0.922 GHz, P = 10 kW, time of MW heating-15 s, $T_o$ = 20 °C. Applied energy = 150 kJ.

### 3.5. Temperature Distribution in Soil Depth

Experiments showed that all applied energy was absorbed by a depth of up to 120 mm in the zones of maximum energy release (90–150 mm from applicator beginning (Figure 16)). In all other zones, the depth of energy penetration was lower.

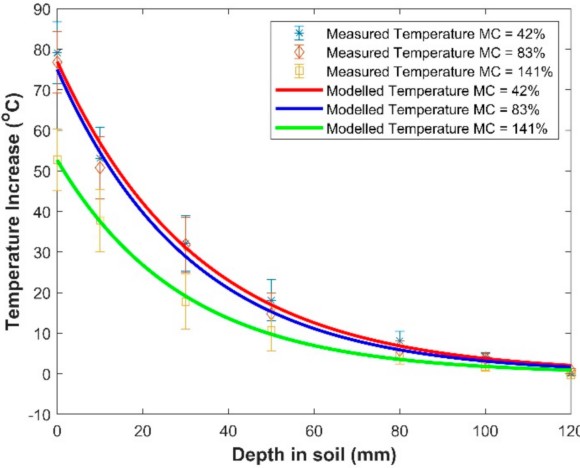

**Figure 16.** Temperature distribution in depth of the wood in applicator central vertical plane on the distance 120 mm from applicator beginning for three moisture contents: MC = 141%, 83% and 42%. (F = 0.922 GHz, applied energy 135–150 kJ) (Error bars represent the standard error for the measured data).

### 3.6. Energy Absorption on the Soil Depth

Table 4 shows the percentage energy distribution as a function of depth and moisture for the Comb 3 applicator. Between 43% and 52% of the applied energy was absorbed in the top 30 mm, 62–71% of the energy was dissipated in the top 50 mm, and all the applied energy was absorbed on the depth up to 120 mm.

**Table 4.** Percentage of the energy absorbed by soil depending on the depth (applicator Comb 3, 0.922 GHz).

| Depth (mm) | MC = 141%, d = 1290 kg m$^{-3}$ | MC = 83%, d = 770 kg m$^{-3}$ | MC = 42%, d = 672 kg m$^{-3}$ |
|---|---|---|---|
| 10 | 19 | 19 | 16 |
| 30 | 52 | 50 | 43 |
| 50 | 74 | 71 | 62 |
| 80 | 96 | 87 | 86 |
| 100 | 99 | 94 | 94 |
| 120 | 100 | 100 | 100 |

Experiments showed that practically all the energy was absorbed by about 270 mm from the beginning of the Comb 3 applicator and 98% of the energy was absorbed by 240 mm from the applicator beginning. The maximum energy release took place at 100–120 mm from the beginning of the applicator.

## 4. Conclusions

Both Comb 1 and Comb 2 give similar energy distributions in the soil along the applicator length, across the applicator width and in soil depth with MC ranging from 32% to 174%. The maximum energy absorption takes place at about 60 mm from applicator beginning. Almost all the energy was absorbed by 200 mm along the applicator and 150 mm on the applicator width for any soil moisture content. The most significant part of the energy was absorbed in the applicator central vertical plane in the top 50 mm of soil. In other volumes of the soil, the share of energy absorbed by surface layers is much higher. The length of the working part of the 2.45 GHz comb applicators was 356 mm. According to these results, the working length of applicators can be reduced to 250 mm without any harm to the energy distribution in the soil.

Most of the energy (up to 95%), irradiated by the 0.922 GHz Comb 3 applicator, was absorbed on the width of about 220 mm across the applicator and differences in the soil moisture content had no significant effect on energy distribution across the applicator. The maximum energy release took place at 100–120 mm from the applicator's beginning. Approximately 71–74% of the energy was absorbed in the applicator's central vertical plane within the top 50 mm of the soil. The length of the working part of the Comb 3 applicator (0.922 GHz) was 346 mm. According to the experimental results, the working length of applicators can be reduced to 300 mm without any harm to energy distribution in the soil.

Mathematical modelling of the temperature distribution in the soil, due to using the various Comb applicators, gives a good approximation to the real situation revealed in these experiments. They can be used for assessment of the temperature distribution from comb applicators of different designs for heating surface layers of soil and other dielectric materials.

Comb applicators provide the following advantage: release of the significant part of applied MW energy in layers close to the applicator surface. It is better compared with a horn irradiator, but this requires quantitative assessment. Comb applicators can provide required soil top layer treatment (sterilization) with reasonable efficiency and can be recommended for practical use in shallow soil treatment for weed seed and pathogen control in agricultural applications.

## 5. Patents

The applicator design presented in this paper is captured in International Publication Number—WO 2018/112531 A1.

**Author Contributions:** Conceptualization, G.B., Y.P. and G.T.; methodology, G.T.; formal analysis, G.B.; investigation, G.T.; resources, G.B.; data curation, G.B.; writing—original draft preparation, G.T. and G.B.; writing—review and editing, G.B.; visualization, G.B.; supervision, G.T.; project administration, G.B.; funding acquisition, G.B. All authors have read and agreed to the published version of the manuscript.

**Funding:** This research was funded by GRAINS RESEARCH AND DEVELOPMENT CORPORATION, grant number UM00053.

**Conflicts of Interest:** The authors declare no conflict of interest. The funders had no role in the design of the study; in the collection, analyses, or interpretation of data; in the writing of the manuscript, or in the decision to publish the results.

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
