# Peer review of "Development of Microwave Slow-Wave Comb Applicators for Soil Treatment at Frequencies 2.45 and 0.922 GHz (Theory, Design, and Experimental Study)"

_agriculture, doi:10.3390/agriculture10120604_

Round 1

Reviewer 1 Report

1. How does the depth of penetration of microwaves into the soil compare with simple radiators such as a horn antenna or an array of horn antennas at the corresponding frequencies of radiation? I expect that both these slow wave combs and the horn antenna will give fairly similar depths of penetration.

2. The authors should also discuss a likely deficiency of the comb applicators in that the power densities will be quite high near the feed point of microwaves but will rapidly decrease as the microwaves progress along the comb structure.

3. I note that the volume of heating with comb applicators is fairly limited. The germination of seeds may be reasonable with such small volumes.However, I do not see applicability of such a limited volume heating system for agricultural field applications. The authors should discuss this issue.

Author Response

Response to Reviewer 1

  1. How does the depth of penetration of microwaves into the soil compare with simple radiators such as a horn antenna or an array of horn antennas at the corresponding frequencies of radiation? I expect that both these slow wave combs and the horn antenna will give fairly similar depths of penetration.

Response: Thank you for the comment. The penetration depth for microwave energy from a horn antenna is directly dependent on the natural attenuation of microwave fields in soil (a). In the case of the medium moisture soil used in this experiment, the natural attenuation of microwave fields with a frequency of 2.45 GHz is estimated to be 3.04 m-1. The slow-wave applicator introduces a much more rapid attenuation of the microwave fields due to its evanescent field response (t2). In the case of the medium moisture soil used in this experiment, the slow-wave attenuation of microwave fields with a frequency of 2.45 GHz is estimated to be 15.7 m-1. This is 5.2 times faster field attenuation than would be expected from a horn antenna. (This has been added to the text of the paper – Line 281 – 288).

  1. The authors should also discuss a likely deficiency of the comb applicators in that the power densities will be quite high near the feed point of microwaves but will rapidly decrease as the microwaves progress along the comb structure.

Response: Thank you for this comment as well. The reviewer is correct. There is attenuation along the length of the comb applicator, which is in accordance with the natural attenuation of the microwave field in the soil (a). This is to be expected and is acceptable because in a practical system the applicator will be pulled slowly over the surface of the soil to provide more uniform treatment along a strip of soil. Although not being reported here, the strategy of moving the applicator has proven to be provide effective long-term control weeds and their seeds in the soil, with some longitudinal experiments showing weed free conditions for more than 200 days (unpublished). (This has been added to the text of the paper – Line 289 – 295).

If reviewer is concerned about opportunity of arcing in the waveguide to comb transition area at high MW power, this problem can be overcome by methods that are commonly used in high power slow wave antennas.

  1. I note that the volume of heating with comb applicators is fairly limited. The germination of seeds may be reasonable with survival in small volumes. However, I do not see applicability of such a limited volume heating system for agricultural field applications. The authors should discuss this issue.

Response: Thank you for this comment as well. As mentioned in the introduction of the paper, most seed banks in minimal tillage systems are located in the top 2 cm of soil; therefore, treatment beyond 2 cm in depth is wasteful of energy, hence the motivation for the development of these applicators. Additionally, as mentioned in our response to the previous comment, unpublished longitudinal experiments have demonstrated effective control (i.e., bare soil) in treatment plots for more than 200 days. The text has not been altered; however, we hope that this adequately addresses the reviewer’s concerns.

Reviewer 2 Report

This article is the extension of the conference abstract AMPERE 2019.

It does not contain sufficient novelty compared to their earlier work DOI: 10.4995/AMPERE2019.2019.9651

Author Response

Response to Reviewer 2

  1. It does not contain sufficient novelty compared to their earlier work DOI: 10.4995/AMPERE2019.2019.9651.

Response: Thank you for this comment. The paper that was presented at the AMPERE conference considered slow-wave and Frustrated Total Internal Reflection (FTIR) applicators, only operating at 2.45 GHz. The earlier paper made no attempt to model the heating profiles of these applicators. This paper considers only slow-wave applicators, but extends the study to include two separate frequencies and extensive modelling of the temperature profile to be expected in the soil. We therefore believe that there is novelty in the current paper that was not available in the previous paper.

Reviewer 3 Report

Dear Authors,

The topic about microwave treatment of soil is interesting and novel theme.

I would like to propose you some improvements of your paper:

  1. Be consistent about the space between numbers and units. sometimes you leave a blank and sometimes not; examples: 2.45 GHz and 0.922GHz (line 22). 
  2. The same applies to different words throughout the text. Sometimes there is no space between words, sometimes there is a double space. Please check.
  3. The applicator design paragraph does not seem to have its place in the INTRODUCTION but rather in MATERIALS AND METHODS (Experimental installations and procedure.
  4. Figure 8 - some of the text is not visible 
  5. Table 3 - Moisture of soil used in relation to Comb 1: a value ~ 80 % seems to be missing. 

I am quite unsure about the operation of a 2.45 GHz 30 kW generator at 3.5 kW. If magnetron operated, please give more details about the type of the generator and magnetron's operation.

Author Response

Response to Reviewer 3

I would like to propose you some improvements of your paper:

  1. Be consistent about the space between numbers and units. sometimes you leave a blank and sometimes not; examples: 2.45 GHz and 0.922GHz (line 22).

Response: Thank you for picking up these problems. The text throughout the paper has been revised and corrected.

  1. The same applies to different words throughout the text. Sometimes there is no space between words, sometimes there is a double space. Please check.

Response: Thank you for picking up these problems. The text has been revised and corrected throughout the paper.

  1. The applicator design paragraph does not seem to have its place in the INTRODUCTION but rather in MATERIALS AND METHODS (Experimental installations and procedure.

Response: Thank you. The Introduction and Method Sections have been modified to suit. (This has been corrected in the paper – Line 127 – 156).

  1. Figure 8 - some of the text is not visible

Response: The image for Figure 8 has been updated.

  1. Table 3 - Moisture of soil used in relation to Comb 1: a value ~ 80 % seems to be missing.

Response: The reviewer is correct. Unfortunately, there was no data gathered for  ̴  80 % moisture content for Comb 1; therefore, this can not be corrected.

I am quite unsure about the operation of a 2.45 GHz 30 kW generator at 3.5 kW. If magnetron operated, please give more details about the type of the generator and magnetron's operation.

Response: Magnetrons can produce variable microwave power output, with variation between approximately 10 % of their rated output and 100 % of their rated output. Below 10 % of the rated output power, the magnetron’s output becomes less stable. In this case the 2.45 GHz magnetron system was operating at 11.7 % of its rated output; therefore, it should be operating within its stable range. The magnetron that was used in the experiment was a Model CWM-30S with an operating power range from 3 to 30 kW. The 60 kW, 0.922 GHz magnetron was operating at between 15 % and 17 % of its rated output, so it was also within its stable range. (This has been corrected in the paper – Line 209 – 215).

Reviewer 4 Report

23 and 330 (table T4) edit kg/m3 to read kg/m3

In Fact there are many instances where this needs to be done in the text.

39 Lambert’s law concerns the transmission of light through a medium. Best state Maxwell’s equations, which for a semi-infinite slab reduces to an exponential decay. Note in 107 the authors describe the soil as a semi-infinite slab.

53 What is an “Impedance electrode”? Authors to explain

94 The authors quote, “The problem is solved iteratively”? Which parameters are solved for using (2) and (3)?

98 Dimension y is not shown in any sketch.  

101 Not at all clear how from (4) one gets to (5)? What is the relationship between y and x?

102 Are the dimensions of (5) correct?

105 A has not been defined in jpy/A

110-115 Not clear how one gets from (6) to (7)

127 Do the authors mean “groove” not “grove”?

131 Why 3 mm? is it because it was available or chosen specifically through the design parameters?

169-170 Was an iso-circulator used in the experimental set up?

215, 226, 240, 249, 275 279, 290-292, 312, edit To so that o is a subscript

229 I do not see any error bars in the experimental temperatures shown on Figure 11 and indeed subsequently in Figures 12 and 16. Have the authors repeated the experiments and found no variation at all?

Author Response

Response to Reviewer 4

23 and 330 (table T4) edit kg/m3 to read kg/m3

Response: These have been corrected – Line 364. Thank you.

39 Lambert's law concerns the transmission of light through a medium. Best state Maxwell's equations, which for a semi-infinite slab reduces to an exponential decay. Note in 107 the authors describe the soil as a semi-infinite slab.

Response: The text has been changed to remove the reference to Lambert's law and reference Maxwell’s equations and propagation in semi-infinite solids – Line 39.

53 What is an "Impedance electrode"? Authors to explain

Response: The term “impedance electrode” has been replaced with “comb structure” – Line 48

94 The authors quote, "The problem is solved iteratively"? Which parameters are solved for using (2) and (3)?

Response: The wording has been revised to indicate that the value of t2 must be solved iteratively – Line 96. Thank you.

98 Dimension y is not shown in any sketch.

Response: Figure 1 has been revised to show the coordinate system for the comb structures. Thank you.

101 Not at all clear how from (4) one gets to (5)? What is the relationship between y and x?

Response: the text has been updated to explain how the equations are related - The temperature distribution is proportional to the square of the microwave field distribution and is determined from this squared field distribution by integration, as outlined by Crank [1]. Line 103 – 105. The coordinates x and y represent different spatial coordinates, as outlined in the revised version of Figure 1.

102 Are the dimensions of (5) correct?

Response: Yes, we understand that the units for equation (5) are correct.

105 A has not been defined in jpy/A

Response: A is already defined in Table 1 as being the width of the slow-wave structure (m).

110-115 Not clear how one gets from (6) to (7)

Response: The microwave heating problem is a thermal diffusion process; therefore, according to Holman [2], the multi-dimensional temperature distribution is determined by multiplying the separate one-dimensional temperature distributions together. The text has been updated to explain this – Line 116 – 119.

127 Do the authors mean "groove" not "grove"?

Response: Thank you. Yes, this has been corrected throughout the paper.

131 Why 3 mm? is it because it was available or chosen specifically through the design parameters?

Response: The ceramic plates affect the microwave attenuation parameter t2, which restricts the microwave heating to the upper layer of the soil. The thickness of the plates was chosen, based on variations in field distributions observed from electromagnetic simulation using the Finite-difference Time-domain simulation processes developed by Yee [3]. A Note has been added at the foot of Table 2 – Line 136 – 139.

169-170 Was an iso-circulator used in the experimental set up?

Response: Yes, an isolator was included in the set up. The text has been altered to reflect this – Line 180.

215, 226, 240, 249, 275 279, 290-292, 312, edit To so that o is a subscript

Response: Thank you. This has been corrected.

229 I do not see any error bars in the experimental temperatures shown on Figure 11 and indeed subsequently in Figures 12 and 16. Have the authors repeated the experiments and found no variation at all?

Response: Thank you. Measured data represents the mean of 3 replicates. Standard Error has been determined and included as error bars on the various figures.

Response to General Comments from Editorial Team

As the repetition rate detected is high, please also revise the manuscript during revision. The iThenticate report is attached below. Please check and revise the manuscript to ensure the repetition rate of single paper is no more than 5%.

Response: Thank you. Some effort to reduce the repetition rate has been made; however, it is very difficult because a good number of the reptations are in definitions of parameters and figure captions etc., which would lose their meaning if the text was altered too much.

  1. Crank, J., The Mathematics of Diffusion. 1979, Bristol: J. W. Arrowsmith Ltd.
  2. Holman, J.P., Heat Transfer. 10th ed. 1997, New York: McGraw-Hill.
  3. Yee, K.S., Numerical solution of initial boundary value problems involving Maxwell's equations in isotropic media. IEEE Transactions on Antennas and Propagation, 1966. 14(3): p. 302-307.

Round 2

Reviewer 4 Report

Equations 4,  5 edit Sin to sin

Line 358 Edit “……….to be provide effective….”

Lines 283, 321, 376, 388, 389, Change To to To
